# Stage Detection of Mild Cognitive Impairment: Region-dependent Graph Representation Learning on Brain Morphable Meshes

**Jiaqi Guo**[*1,2]                                          JIAQI.GUO@NORTHWESTERN.EDU

**Emanuel A. Azcona**[1,2]                      EMANUELAZCONA@U.NORTHWESTERN.EDU

**Santiago López-Tapia**[1,2]                  SANTIAGO.LOPEZTAPIA@NORTHWESTERN.EDU

**Aggelos K. Katsaggelos**[1,2]                   KATSAGGELOS@NORTHWESTERN.EDU

[1] *Image and Video Processing Lab (IVPL), Northwestern University, IL, USA*

[1] *Department of Electrical and Computer Engineering, Northwestern University, IL, USA*

**Editors:** Accepted for publication at MIDL 2023

## Abstract

Mild cognitive impairment (MCI), as a transitional state between normal cognition and Alzheimer's disease (AD), is crucial for taking preventive interventions in order to slow down AD progression. Given the high relevance of brain atrophy and the neurodegeneration process of AD, we propose a novel mesh-based pooling module, REGIONPOOL, to investigate the morphological changes in brain shape regionally. We then present a geometric deep learning framework with the REGIONPOOL and graph attention convolutions, to perform binary classification on MCI subtypes (EMCI/LMCI). Our model does not require feature engineering and relies only on the relevant geometric information of T1-weighted magnetic resonance imaging (MRI) signals. Our evaluation reveals the state-of-the-art classification capabilities of our network and shows that current empirically derived MCI subtypes cannot identify heterogeneous patterns of cortical atrophy at the MCI stage. The class activation maps (CAMs) generated from the correct predictions provide additional visual evidence for our model's decisions and are consistent with the atrophy patterns reported by the relevant literature.

**Keywords:** deep learning on meshes, graph neural networks, shape analysis, neuroanatomy

## 1. Introduction

Mild cognitive impairment (MCI) represents the transitional state between normal aging and early dementia. It is recognized as an important sign of Alzheimer's disease (AD) since one-third of MCI develop into AD within five years of follow-up. MCI subjects are subdivided by the Alzheimer's disease neuroimaging initiative (ADNI) into two subtypes—early MCI (EMCI) and late MCI (LMCI)—based on the WMS-R Logical Memory II Story A score, a detailed diagnosis criterion can be found at Initiative (2008). Compared with LMCI, subjects in the stage of EMCI demonstrate milder degrees of cognitive and functional impairment and slower disease progression (Aisen et al., 2010), which is considered the optimal stage of applying early therapeutic interventions to reduce the number of AD patients. Recently, there has been a surge of interest in identifying the subtle variations between MCI subtypes, but this is challenging because EMCI and LMCI are classified by a single memory

---

* Corresponding author

score, leading to low specificity or even misclassification. To address this issue, we seek a more objective method that would allow researchers to make a reliable classification of MCI subtypes. Several machine learning (ML) based studies have applied neuroimaging data from ADNI to classify MCI subtypes. Both Nozadi et al. (2018) and Gray et al. (2012) use PET images (positron emission tomography) for MCI classification, while Gray et al. further improve the classification result by combining the PET scan from multiple time points. Sheng et al. (2019) classify the MCI subtypes using the topological organization of brain networks. Shi and Liu (2020) parcelled the fMRI images into multiple regions of interest (ROIs) and extracted features from the average RS-fMRI signal for classification. However, these approaches rely heavily on manual feature extraction, which makes them not only expensive to train but also difficult to interpret the classification results. A comparison of these models will be presented in table 2($b$).

Brain atrophy assessed on structural magnetic resonance imaging (MRI) has been identified as a valid marker of AD-related neurodegeneration (Whitwell et al., 2012). For example, hippocampus atrophy is long regarded as one of the best established and validated biomarkers of AD (Jack Jr et al., 2011), with the progressive 15-40% decrease of the hippocampus volume across the entire disease spectrum according to post-mortem studies (Bosscher and Scheltens, 2002). Likewise, the amygdala atrophy in AD is also prominent (Scott et al., 1991) and closely correlated to the symptom severity in AD and MCI (Poulin et al., 2011; Yi et al., 2016). Besides, Tondelli et al. (2012) indicate that the morphological changes of the brain in shapes even begin ten years prior to the clinical symptoms of AD occurring, which further emphasizes the importance of brain shape analysis in the early detection of AD. In our study, we use 3D morphable triangular mesh manifolds with the same connectivity to approximate the morphology of both cortex and subcortical structures and employ graph neural networks (GNNs) to study their powerful shape-description ability. However, the large scale of the brain meshes presents us with a great challenge, i.e., how to aggregate the vertex embeddings to obtain a descriptive mesh representation while maintaining the information integrity. Many works (Hamilton et al., 2017; Xu et al., 2018) in the literature demonstrate that the choice of *readout functions* contributes significantly to the representational power and performance of the model. To that end, a popular choice is using mesh simplification (Ranjan et al., 2018; Yuan et al., 2020) as a readout function to aggregate vertex embeddings to represent the entire mesh. Based on Ranjan's method, Yuan et al. average the embedding of decimated vertices to ensure the integrity of information, but they failed to address the relative geometric importance among vertices. One major drawback of this kind of method is, considering the computation complexity of the simplification process, one has to compute a down-sampling matrix in advance, leading the down-sampling process to be always deterministic during the training process. Gopinath et al. (2020) proposed a learnable readout function by predicting task-specific pooling regions directly on the mesh surface. But this method is susceptible to different meshing procedures and suffers from low partitioning accuracy, making it not applicable.

Given the high relevance of brain shape in the neurodegeneration process of AD, we propose a geometric deep learning framework built upon the graph attentional convolutions (Veličković et al., 2017) and REGIONPOOL defined in Sec. 2.3, to investigate the morphological changes in brain shape. Similar to the spatial pooling operation in CNNs, our approach REGIONPOOL defines hierarchical "patches" with varying topologies and sizes

on the mesh surface and generates region-level representations by global mean-pooling the embeddings of a set of sampled geometric-important vertices within each region. As we will show later, our proposed model can not only achieve superior performance in EMCI/LMCI classification but can also identify the subtle morphological changes in brain shape and provide high-resolution 3D visual interpretations of *classifier reasoning*, which is a highly desired property of deep learning model in medical image analysis.

## 2. Methods

### 2.1. Graph Construction

All data used in our study are transformed into triangular mesh manifolds and registered to a standard template (same number of vertices/edges per sample) following the procedures in Appendix B. A single mesh instance will contain 47,616 vertices, including 32,768 (69%) from the cortical structure and 14,848 (31%) from the subcortical structure. The feature vector of the cortical vertices has six dimensions corresponding to the Cartesian coordinates of both white matter and gray matter vertices in their subjects' native space. Since all corresponding edges of white and gray matter share the same edge weight $\mathbf{e}_{i,j}$, they can be considered to form the same "faces" with different coordinates (Azcona et al., 2020). Similarly, subcortical vertices are assigned three features, which is also the Cartesian coordinate of the subcortical vertices in their subjects' native space. As shown in Figure 1, according to the anatomy structure, the mesh vertices are categorized into four types— Rh-Cortex/Subcortical, Lh-Cortex/Subcortical vertices and associated by two relations— Cortex/Subcortical connections.

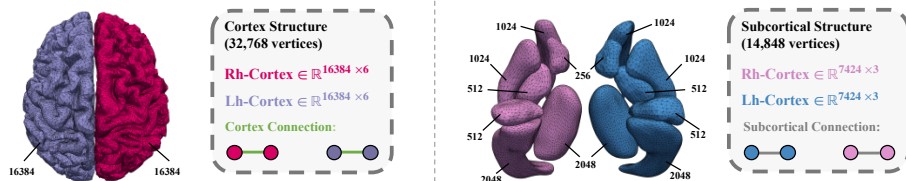

Figure 1: Graphical illustration of our constructed heterogeneous mesh.

### 2.2. Heterogeneous Brain Mesh Attention Network (Het-BMAT)

As illustrated in Fig. 2, our proposed network, Het-BMAT, is built upon the GATconv (Veličković et al., 2017), REGIONPOOL in Sec. 2.3, and a combination of hetero and normal linear layers. In addition, the heterogeneity of our model will enable information to propagate between different vertices through different relations (Appendix A, definition A.2).

As an initial step, a standard GATconv is used as the "pre-conv" block to map the input features to a high dimensional space to obtain sufficient expressive power. Then, given the different sizes of substructures of the brain meshes, we propose a simple mesh attention (MAT) block to provide various receptive fields, addressing the "over-smoothing" problem by introducing skip connection (Veit et al., 2016) to every stacked GATconv layer. In the following experiment, all GATconvs will share the same channel size of 16. Besides, to get full use of the *inherent coarse-grained hierarchies* of the brain meshes, we adopt two

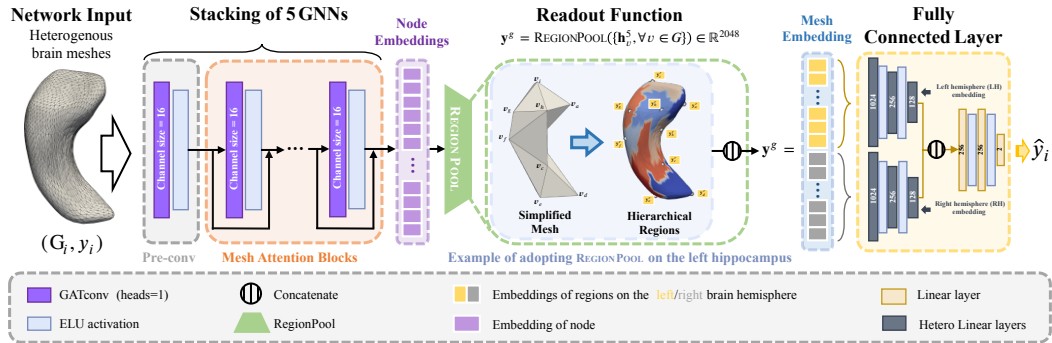

Figure 2: Our network architecture. Given a 3D mesh input, $G_i$, the Het-BMAT network will predict its corresponding label, $y_i$, from the learned mesh representation, $y^g$.

hetero linear layers to further aggregate the output of REGIONPOOL (i.e., a concatenation of region-level embedding) to a higher-level embedding space—the embeddings of the *left and right hemispheres*. Finally, an exponential linear unit (Clevert et al., 2015) is introduced after every convolution and linear layer to add nonlinearity to the network training. We use a softmax function to compute the corresponding predicted possibility and employ a standard binary cross entropy (BCE) loss function to train the model.

## 2.3. Region-dependent Pooling on Mesh Surface

### 2.3.1. Vertices selection via shape-preserving mesh simplification

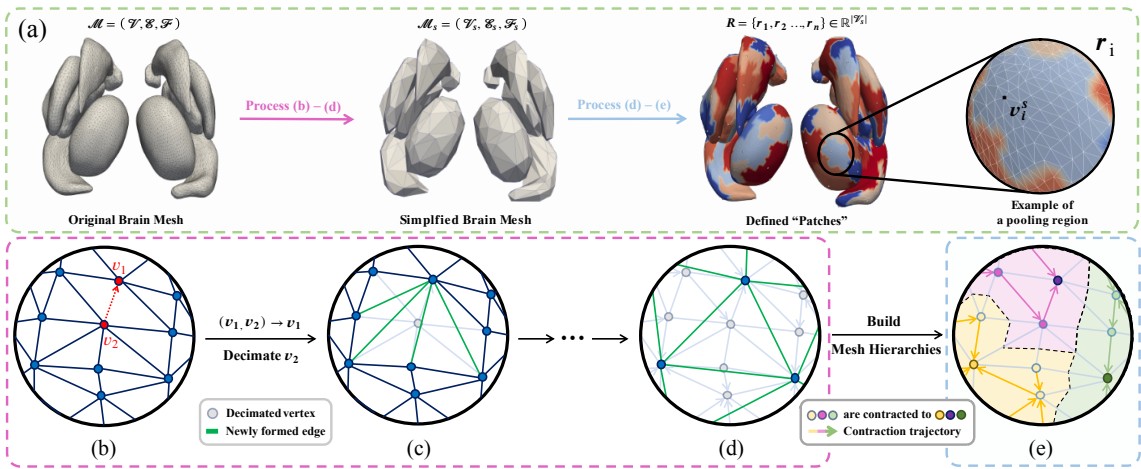

Figure 3: The process of mesh simplification and building the mesh hierarchies. (a) provides the overall pooling pipeline. (b)-(d) illustrate the process of edge contraction via vertex decimation. (d)-(e) illustrate the process of defining hierarchical regions on the original mesh surface based on the simplification process.

Our mesh simplification process is based on the work of (Garland and Heckbert, 1997), which simplifies the mesh by repeatedly contracting the edge with the least *collapse cost*. To begin with, given a plane $\mathbf{p} = [a \ b \ c \ d]^\top$ defined by the equation $ax + by + cz + d = 0$ ($a^2 + b^2 + c^2 = 1$). The *fundamental error quadric* matrix can be written as $\mathbf{K_p} = \mathbf{pp}^\top$, which can be thought of as the quadratic representation of plane $\mathbf{p}$. In the original work, Garland and Heckbert associate each vertex $v_i = [x_i \ y_i \ z_i \ 1]^\top$ with a set of planes $\mathcal{P}$ by defining an error s.t. $\Delta(v) = v^\top Q_\mathcal{P} v$, where $x_i, y_i, z_i$ are the corresponding Cartesian coordinate of $v_i$ and $Q_\mathcal{P}$ is the sum of the corresponding fundamental error quadrics of the planes in $\mathcal{P}$, s.t. $Q_\mathcal{P} = \sum_{\mathbf{p} \in \mathcal{P}} \mathbf{K_p}$. For a given edge contraction $(v_1, v_2) \to \bar{v}$, they use $\bar{Q} = Q_i + Q_j$ as a new error quadric to approximate the error at the target contraction position $\bar{v}$. Note that in our study, we choose a simple scheme to decide the position of $\bar{v}$, i.e., to select either $v_1$ or $v_2$ depending on which one of them produces a smaller *decimation error* $\Delta(\bar{v})$. To better illustrate, given the condition $\Delta(v_1) < \Delta(v_2)$, we will have: $\omega_2 = \Delta(v_1) = \min(\Delta(v_1), \Delta(v_2))$. Here, we use $\omega_2$ to denotes the corresponding collapse cost of contracting the edge $(v_1, v_2)$ to $v_1$ by decimating $v_2$. A graphical example of this process can be found in Fig. 3b to 3c, while Fig. 3d provides an example of the simplified mesh surface, with the grey arrows indicating the trajectories of edge contraction.

### 2.3.2. COST-ORIENTED SAMPLING

To address the issues mentioned in Sec 1 and obtain a descriptive mesh representation. Our REGIONPOOL further exploits this simplification process by adopting a cost-oriented sampling strategy. The key intuition here is that we want our model to learn the mesh representation not only from the vertices of the simplified mesh, $\mathcal{V}_s$, but also from those decimated vertices, $\mathcal{V}_d$. For this purpose, we first define hierarchical "patches" with varying *topologies* and *sizes* on the mesh surface (Fig. 3d to 3e). As mentioned in the previous section, at each step of the mesh simplification, we contract a valid edge $(v_i, v_j)$ by selecting the target position $\bar{v}$ from either $v_i$ or $v_j$ depending on their corresponding decimation error $\Delta$. Thus, by simplifying meshes in such a one-to-one manner, the vertices of the simplified mesh, $\mathcal{V}^s = \{v_1^s, v_2^s, ..., v_n^s\} \in \mathbb{R}^n$, and decimated vertices, $\mathcal{V}^d = \mathcal{V} \setminus \mathcal{V}^s$, will eventually form $n$ independent *pooling regions*, denoted as $R = \{r_1, r_2..., r_n\} \in \mathbb{R}^n$. As shown in Fig. 3a, any region $r_i$ consists of one vertex from the simplified mesh, $v_i^s \in \mathcal{V}^s$, and a set of decimated vertices, $\mathcal{V}_i^d \subset \mathcal{V}^d$, that were iteratively contracted to $v_i^s$ during the mesh simplification process.

Next, we propose a cost-oriented sampling strategy for all $r_i \in R$, providing a self-adaptive approach to extract the most representative features from every mesh hierarchy, i.e., for a given region $r_i$, we defined the corresponding sampling strategy $S_i(\cdot)$ to be a random sample (without replacement) of $m$ decimated vertices from a specific categorical distribution, $Cat(\pi_1, \pi_1, ..., \pi_k)$. Here, the class probability $\pi_k$ indicates the probability of $v_k^d$ being sampled, which is defined as: $\pi_k = \frac{\omega_k}{\sum_{k=1}^{|\mathcal{V}_i^d|} \omega_k}$, $\forall k \in \{1, 2, ..., |\mathcal{V}_i^d|\}$ where $\omega_k$ is the corresponding collapse cost of the decimated vertex $v_k^d$, defined in equations 2.3.1, considering the geometric significance of $\omega_k$ (i.e., the geometric error caused by contracting a certain vertex to a new position), we normalize it and interpret it as the relative importance of $v_k^d$. Note that if the number of samples exceeds the number of decimated vertices in that region, i.e., $m > |\mathcal{V}_i^d|$, we will directly take all decimated vertices to be the sampling result,

i.e., $S_i(\mathcal{V}_i^d) = \mathcal{V}_i^d$. By selecting an appropriate number of sampling, this process allows our model always to be able to learn from a group of vertices with the highest geometric importance in every patch, ensuring the stability of results. Besides, it also prohibits the loss of localized information during the global edge contraction since $Cat(\cdot)$ are measured from the local importance within a single patch. A toy example of this process can be found in Appendix C.1.2

Following that, we compute the region-level embedding, $\mathbf{y}_i^r \in \mathbb{R}^{c^{(K)}}$, by element-wise pooling the embeddings of both sampled vertices $v_i^s$ and $S_i(\mathcal{V}_i^d) \subset \mathcal{V}_i^d$. Finally, those region-level embeddings are concatenated as the representation of the entire mesh s.t.

$$\mathbf{y}^g = \sigma \left( \overset{n}{\underset{i=1}{\|}} \text{Pool} \left( \mathbf{h}_{v_i^s}^{(K)}, \left\{ \mathbf{h}_{v_i^d}^{(K)}, \ \forall v_i^d \in S_i(\mathcal{V}_i^d) \right\} \right) \right) \tag{1}$$

where $\sigma$ is any possible non-linear activation function (e.g., ReLU, ELU), and Pool can be any regular pooling method (e.g., global max/mean/sum/attention-pooling). By default, we choose global mean-pooling to support the Grad-CAM (Selvaraju et al., 2017) visualization in Sec. 3.4 and take ELU as the activation function.

## 3. Experiment

### 3.1. Dataset and experiment Setting

In this paper, we use 4,072 T1-weighted MRI volumes from the Alzheimer's Disease Neuroimaging Initiative (ADNI) dataset, which are the longitudinal data collected over several years on 492 individual subjects. Basic demographic information includes the number of samples with different labels, as well as the age and gender distribution of the subjects are shown in Table 4. Our network was trained with 16 samples per batch over 200 epochs and adopted an Adam (Kingma and Ba, 2014) optimizer with a learning rate of $1 \times 10^{-3}$ and a decay rate of 0.95 per 5 epochs to minimize a BCE loss function. Besides, the weights of our network were initialized using Xavier uniform initializer (Glorot and Bengio, 2010) and $L_2$-regularized (weight decay) with a weight decay of $1 \times 10^{-3}$ to avoid over-fitting.

### 3.2. MCI subtypes classification

In our first experiment, we perform 5-fold cross-validation to analyze the effectiveness of incorporating different brain structures, i.e., the cortex/subcortical/both structures, for MCI subtypes classification and report the averaged result. Note that to avoid samples from the same subjects appearing in both training and testing sets, the data samples are shuffled with respect to their subject identifiers. Following the convention defined in Sec. 2.3, we respectively divide the subcortical and cortex structure into $n = 128$ pooling regions, and for every single region $r_i$, we will sample $|\mathcal{V}_i^d| = 40$ decimated vertices to do the RegionPool. See Appendix C.1.1 for a detailed description of parameter selection.

Table 1 summarizes the results of our ablation study with different brain structural inputs. Our Het-BMAT network performs best by incorporating the subcortical structures as input. Although ordered atrophy patterns in MCI patients have been proven to be associated with their profiles of increasing cognitive dysfunction, the classifier's performance does not improve with the inclusion of cortex structures. It becomes even worse when only

Table 1: 5-fold cross-validation of EMCI vs. LMCI classification using different brain structures

| Structure | Threshold=0.5 | | | AUC | |
|---|---|---|---|---|---|
| | Precision | Recall | F1 | ROC-AUC | PR-AUC |
| all structures | $0.823 \pm 0.027$ | $0.810 \pm 0.047$ | $0.812 \pm 0.044$ | $0.877 \pm 0.040$ | $0.859 \pm 0.053$ |
| subcortical structures | $\mathbf{0.826 \pm 0.039}$ | $\mathbf{0.822 \pm 0.045}$ | $\mathbf{0.822 \pm 0.044}$ | $\mathbf{0.888 \pm 0.036}$ | $\mathbf{0.865 \pm 0.061}$ |
| cortical structures | $0.667 \pm 0.043$ | $0.651 \pm 0.044$ | $0.661 \pm 0.054$ | $0.689 \pm 0.053$ | $0.650 \pm 0.095$ |

[1] All results in this paper are demonstrated in the form of $mean \pm std$ (across folds)
[2] Precision, Recall, and F1-score in this paper are weighted averaged
[3] *ROC/PR-AUC* Area Under the Receiver Operating Characteristic/Precision-Recall Curve

cortex structures are used as input, suggesting that the morphological changes occurring in the subcortical structures are more critical for the EMCI/LMCI classification. Moreover, it also indicates that current empirically derived MCI subtypes (based on the subject's neuropsychological profile) may not be able to capture the heterogeneity of cortical atrophy at the MCI stage (Edmonds et al., 2016).

### 3.3. Baseline comparison

Our REGIONPOOL acts as a readout/global-pooling module in our proposed model, which contributes significantly to the model's representation power and performance. In this section, based on the fundamental architecture of the Het-BMAT network, we evaluate our proposed REGIONPOOL against seven baseline pooling modules on the same classification task in Sec. 3.2: including six commonly used *global-pooling* approaches for graph-level classification, i.e. Global Mean/Add (Lin et al., 2013)/Attention (Li et al., 2015) Pooling, Set2Set (Vinyals et al., 2015), SortPool (Zhang et al., 2018), Graph Multiset Transformer (Baek et al., 2021), and one *hierarchical pooling* approach, i.e. MESHSIM (Ranjan et al., 2018). All experiments were conducted according to the setup in Appendix C.2, perform 5-fold cross-validation and use only subcortical structures as input. Table 2(a) demonstrates that our REGIONPOOL outperforms all the baseline pooling modules on all metrics. This proves that our approach can indeed leverage the information embedded in the graph hierarchical structure and the decimated vertices to improve the overall descriptive ability of mesh representation.

Table 2: Baseline comparison of several graph-based pooling modules and prior studies on EMCI vs. LMCI classification

(a) REGIONPOOL vs. Graph pooling approaches

| Module | Threshold=0.5 | | | AUC | |
|---|---|---|---|---|---|
| | Precision | Recall | F1 | ROC-AUC | PR-AUC |
| **RegionPool** | $\mathbf{0.826 \pm 0.039}$ | $\mathbf{0.822 \pm 0.045}$ | $\mathbf{0.822 \pm 0.044}$ | $\mathbf{0.888 \pm 0.036}$ | $\mathbf{0.865 \pm 0.061}$ |
| MESHSIM | $0.812 \pm 0.036$ | $0.811 \pm 0.035$ | $0.810 \pm 0.035$ | $0.880 \pm 0.032$ | $0.857 \pm 0.057$ |
| GAP | $0.763 \pm 0.022$ | $0.760 \pm 0.023$ | $0.759 \pm 0.022$ | $0.815 \pm 0.031$ | $0.803 \pm 0.053$ |
| GMP | $0.773 \pm 0.020$ | $0.770 \pm 0.018$ | $0.770 \pm 0.017$ | $0.803 \pm 0.025$ | $0.785 \pm 0.038$ |
| GAT | $0.768 \pm 0.017$ | $0.766 \pm 0.019$ | $0.766 \pm 0.019$ | $0.829 \pm 0.021$ | $0.801 \pm 0.048$ |
| SET2SET | $0.771 \pm 0.017$ | $0.771 \pm 0.017$ | $0.770 \pm 0.017$ | $0.832 \pm 0.025$ | $0.809 \pm 0.04$ |
| SORTPOOL | $0.728 \pm 0.013$ | $0.726 \pm 0.013$ | $0.722 \pm 0.016$ | $0.830 \pm 0.022$ | $0.803 \pm 0.049$ |
| GMT | $0.767 \pm 0.017$ | $0.766 \pm 0.018$ | $0.766 \pm 0.017$ | $0.829 \pm 0.020$ | $0.802 \pm 0.049$ |

[1] Method introduction and implementation details are provided in Appendix C.2
[2] Gap/Gmp/Gat global add/mean/attention pooling. Gmt graph multiset transformer.
[3] The baseline models are implemented based on *Pytorch Geometric 2.0.4* (Fey, 2022)

(b) Het-BMAT vs. Related studies

| Study | Target (EMCI/LMCI) | Method | Classifier | Acc |
|---|---|---|---|---|
| **Ours (2022)** | **259/233** | Learn shape descriptors of neuroanatomical structures | GNN | 0.826 |
| Goryawala et al. (2015) | 114/91 | Use both cortical volume & neuropsychological scores | LDA | 0.736 |
| Jie et al. (2018) | 56/43 | Combine temporal & spat-ial properties of DCNs | SVM | 0.787 |
| Nozadi et al. (2018) | 164/189 | Extract regional features from FDG/AV45-PET | RF | 0.725 |
| Shi and Liu (2020) | 77/64 | Extract features from rs-fMRI signals using HHT | SVM | **0.879** |

[1] *LDA* linear discriminant analysis. *DCNs* dynamic connectivity networks. *SVM* support vector machine. *RF* random forest. *HHT* Hilbert-Huang transform.

Apart from that, we also compare our Het-BMAT network against a number of prior studies on the same classification task. We only select studies with a subject number close to 100. As shown in Table 2(b), our proposed model not only achieves comparable and

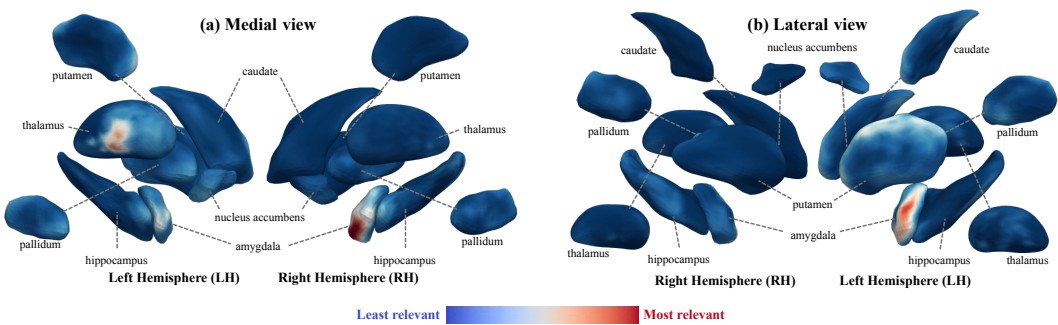

Figure 4: Grad-Cam visualization for all correct predictions by the Het-BMAT network. (a) and (b) are, respectively, the lateral and medial views of the CAM projected on the mesh template.

promising results but also possesses the following merits. *Simple process:* Our approach does not require any feature engineering. *More reliable:* Our model is trained on a larger dataset. *More objective:* The classification is completely based on the geometric information of brain neuroanatomical structures from the MRI signals.

### 3.4. Gradient-weighted class activation mapping (Grad-CAM) visualization

Following the convention defined in (Selvaraju et al., 2017), we employ a Het-BMAT model pre-trained on subcortical structures to generate class activate maps (CAMs) for each correctly classified sample (TP & TN predictions) in the test set. Those CAMs are then averaged and projected onto the subcortical template surface, which is shown in Fig. 4, where we can observe a significant involvement of the amygdala, left putamen, and left thalamus shape in the EMCI/LMCI classification, which matches the findings in (de Jong et al., 2008; Yi et al., 2016). Besides, we also observe a remarkable asymmetry in CAMs, i.e., compared with the RH, the morphological change in LH is more pronounced and indicative. Such left lateralization of brain atrophy in AD progression has been reported by (Long et al., 2013; Thompson et al., 2007) and is thought to be related to the linguistic nature of neuropsychological tests in AD diagnosis (Keilp et al., 1996).

### 4. Conclusion

Our study investigates the corresponding cortical and subcortical atrophy patterns in MCI subtypes (EMCI/LMCI) by learning representations of the brain meshes regionally. The ablation study reveals the limitation of using a single neuropsychological score to capture the heterogeneous patterns of cortical atrophy at the MCI stage in the diagnosis of AD, suggesting the need to incorporate diverse profiles into current cognition-based diagnostic criteria for more meaningful staging of MCI subtypes. Notably, our Het-BMAT network together with REGIONPOOL demonstrates a superior MCI subtype classification capability using only the geometric information of subcortical structures and outperforms the state-of-the-art methods in the baseline comparison. In addition, the generated class activation maps (CAMs) provide qualitatively visual interpretations of classifier decisions and are consistent with the morphological changes reported in the related literature, providing solid evidence for the feasibility of employing geometrical deep learning in disease diagnosis.

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

## Appendix A. Supplementary Definition

In this section, we define some important terminologies related to the heterogeneous graphs used in our study. A detailed description of all mentioned notations in this paper is summarized in Table 3 for quick reference.

**Definition A.1. Triangular Mesh Manifolds.** In graph signal processing (Wu et al., 2020), meshes can be treated as a special variation of the undirected graph and defined as $\mathcal{M} = (\mathcal{V}, \mathcal{E}, \mathcal{F})$, where $\mathcal{V}$ is a finite set of $|\mathcal{V}| = p$ vertices, and $\mathcal{E}$, $\mathcal{F}$ are respectively a set

Table 3: Description of notations in this paper

| Notations | Descriptions |
|---|---|
| $\mathbb{R}^N$ | N-dimensional Euclidean space |
| $p, q$ | The number of nodes and edges in a graph |
| $c, d$ | The dimension of the node and edge feature vector |
| $n$ | The number of vertices in the simplified mesh |
| $R \in \mathbb{R}^n$ | A set of all pooling regions, defined in Sec. 2.3.1 |
| $r_i$ | A single pooling region, where $r_i \in R$ |
| $\mathcal{N}_i$ | The neighborhood nodes set of $v_i$ |
| $\mathbf{x}_i \in \mathbb{R}^c$ | The vertex feature vector of $v_i$ |
| $\mathbf{X} \in \mathbb{R}^{p \times c}$ | The vertex feature matrix |
| $\mathbf{e}_{i,j} \in \mathbb{R}^d$ | The edge feature vector of $(v_i, v_j)$ |
| $\mathbf{E} \in \mathbb{R}^{q \times d}$ | The edge feature matrix |
| $\mathbf{h}_i^{(k)}$ | Hidden state of node $i$ at $k$-th GNN layer |
| $\mathbf{y}^g$ | The embedding of the entire graph $G$ |
| $\mathbf{y}_i^r$ | The embedding of the pooling region $r_i$ |
| $(v_i, v_j) \to \bar{v}$ | Contracting edge $(v_i, v_j)$ into $\bar{v}$ via vertex decimation |
| $\mathcal{V}^s \in \mathbb{R}^n$ | A set of vertices in the simplified mesh |
| $\mathcal{V}^d \in \mathbb{R}^{p-n}$ | A set of all decimated vertices, where $\mathcal{V} = \mathcal{V}^s \cup \mathcal{V}^d$ |
| $\mathcal{V}_i^d \in \mathbb{R}^{m_i}$ | A set of decimated vertices that being contracted to $v_i^s$ |
| $\mathcal{A}, \mathcal{R}$ | Vertex types and relations in the heterogeneous graphs |
| $\sigma(\cdot)$ | Non-linearity activation function |
| $\Theta, \gamma, \psi$ | Function with learnable parameters |
| $S_i(\cdot)$ | The cost-oriented sampling defined for pooling region $r_i$ |
| $m$ | The sample size of the cost-oriented sampling |
| $\|$ | Vector concatenation |

of edges and faces. Each vertex in the mesh is assigned with a $c$-dimensional feature vector $\mathbf{x}_i \in \mathbb{R}^c$, and a vertex feature matrix $\mathbf{X} \in \mathbb{R}^{p \times c}$ is used to encapsulate all those feature vectors. Similarly, a $d$-dimensional feature vector $\mathbf{e}_{i,j} \in \mathbb{R}^d$ is also defined for the vertex connection $(v_i, v_j)$ between $v_i$ and $v_j$ and will be stored in the edge feature matrix $\mathbf{E} \in \mathbb{R}^{q \times d}$.

**Definition A.2. Heterogeneous Graph (Sun and Han, 2013).** "A heterogeneous graph, denoted as $\mathcal{G} = (\mathcal{V}, \mathcal{E})$, consists of an object set $\mathcal{V}$ and a link set $\mathcal{E}$. A heterogeneous graph is also associated with a node type mapping function $\phi : \mathcal{V} \to \mathcal{A}$ and a link type mapping function $\psi : \mathcal{E} \to \mathcal{R}$. $\mathcal{A}$ and $\mathcal{R}$ denote the sets of predefined object types and link types, where $|\mathcal{A}| + |\mathcal{R}| > 2$."

**Definition A.3. Graph-level classification with GNN.** For a set of labeled graphs, $\{(G_1, y_1), (G_2, y_2), ..., (G_i, y_i)\}$, where $y_i \in \mathcal{Y}$ is the corresponding label of $G_i \in \mathcal{G}$. The ultimate goal of graph-level classification is to learn the mapping $f$ from $\mathcal{G}$ to $\mathcal{Y}$ (i.e.,

$f : \mathcal{G} \to \mathcal{Y}$). A major challenge of applying GNNs to graph-level classification is to derive a general representation of a given graph $G$ from node embeddings of the $K$-th layer via a readout function.

$$\mathbf{y}^g = \text{Readout}\left(\left\{\mathbf{h}_v^{(K)}, \forall v \in G\right\}\right) \tag{2}$$

A commonly used approach for readout functions is to apply global max/mean/sum pooling to the embeddings of all nodes in a graph. While this technique performs well on "small" graphs, it falls short when dealing with graphs of "large" scale due to the information loss that occurs during dimension compression.

**Definition A.4. Message Passing Scheme (Gilmer et al., 2017).** Message Passing Neural Network (MPNN) defines convolution directly on the graph and operates on a set of spatially closed neighborhood vertices. Generally, message passing consists of two steps, i.e., *message computation* and *aggregation*. Consider the message passing at vertex $v_i$, the message computation operation takes the embedding of the neighborhood vertices of $v_i$ (including $v_i$) and the corresponding edge feature $\mathbf{e}_{i,j} \in \mathbf{R}^d$ (optional) at the previous layer, creating a new message $\mathbf{m}_j^{(k)}$ by applying a certain transformation $\psi^{(k)}$, such that:

$$\mathbf{m}_j^{(k)} = \psi^{(k)}\left(\mathbf{h}_i^{(k-1)}, \mathbf{h}_j^{(k-1)}, \mathbf{e}_{i,j}^{(k-1)}\right), \; \forall j \in \mathcal{N}_i \tag{3}$$

The message aggregation operation aggregates the transformed messages from the neighborhood vertices of $v_i$ and passes it through a function $\gamma^{(k)}$ with learnable parameters (e.g., Multi-Layer Perceptrons (MLPs) (Fey and Lenssen, 2019)). Therefore, a single message passing operator is defined as:

$$\mathbf{h}_i^{(k)} = \gamma^{(k)}\left(\mathbf{h}_i^{(k-1)}, \underset{j \in \mathcal{N}_i}{\Box}\, \psi^{(k)}\left(\mathbf{h}_i^{(k-1)}, \mathbf{h}_j^{(k-1)}, \mathbf{e}_{i,j}^{(k-1)}\right)\right) \tag{4}$$

where $\Box$ represents differentiable permutation-invariant operation (e.g., $\Sigma$, Max, Mean).

**Definition A.5. Graph Attention Convolution (Veličković et al., 2017).** Based on the message passing scheme defined in Sec. A, Veličković et al. (2017) introduces a shared attention mechanism $\mathbf{a}$ on every pair of vertices $(v_i, v_j)$ to compute the attention weight $\alpha_{i,j}$, which indicates the relative/normalized importance of $j$'s features to $i$. The graph attentional convolution (GATconv) can be typically defined as: $\mathbf{h}_i^{(k)} = \alpha_{i,i}\Theta\mathbf{h}_i^{(k-1)} + \sum_{j \in \mathcal{N}_i} \alpha_{i,j}\Theta\mathbf{h}_j^{(k-1)}$. Here, consider the multi-dimensional edge features $\mathbf{e}_{i,j}$ of graphs, the attention weight $\alpha_{i,j}$ are specially computed as:

$$\alpha_{i,j} = \frac{\exp\left(\text{LeakyReLU}\left(\mathbf{a}^\top\left[\mathbf{\Theta}^{(k)}\mathbf{h}_i^{(k-1)} \,\middle\|\, \mathbf{\Theta}^{(k)}\mathbf{h}_j^{(k-1)} \,\middle\|\, \mathbf{\Theta}_e^{(k)}\mathbf{e}_{i,j}^{(k-1)}\right]\right)\right)}{\sum_{k \in \mathcal{N}_i \cup \{i\}} \exp\left(\text{LeakyReLU}\left(\mathbf{a}^\top\left[\mathbf{\Theta}^{(k)}\mathbf{h}_i^{(k-1)} \,\middle\|\, \mathbf{\Theta}^{(k)}\mathbf{h}_k^{(k-1)} \,\middle\|\, \Theta_e^{(k)}\mathbf{e}_{i,k}^{(k-1)}\right]\right)\right)} \tag{5}$$

where $\mathbf{a}$ is a single-layer feedforward neural network. The weight matrices $\Theta_e \in \mathbb{R}^{d^{(k)} \times d^{(k-1)}}$ and $\Theta \in \mathbb{R}^{c^{(k)} \times c^{(k-1)}}$ denote the corresponding linear transformations that map the input features of edges and vertices to a higher-level space.

## Appendix B. Description of Dataset

The data utilized in this study were sourced from the Alzheimer's Disease Neuroimaging Initiative (ADNI) database (adni.loni.usc.edu). Initiated in 2004, ADNI is a longitudinal multi-center study led by Dr. Michael W. Weiner, with the objective of detecting and tracking the progression of Alzheimer's disease (AD) in its early stages through the analysis of various clinical, imaging (e.g., MRI, PET image data), genetic, and biochemical markers.

### B.1. Mesh Extraction

Starting from the T1-weighted MRIs, each scanned image was processed by a set of preprocessing procedures, including image denoising using FreeSurfer v6.0 (Bruce, 2012), B1 field homogeneity correction, and intensity/spatial normalization to extract the wanted surfaces. For the cortex structure, the inner cortical surfaces (modeling the interface between grey and white matter) and outer cortical surfaces (modeling the cerebrospinal fluid/grey matter interface) were extracted. Besides, seven subcortical structures (amygdala, nucleus accumbens, caudate, hippocampus, pallidum, putamen, thalamus) per hemisphere were segmented and modeled into the surface using SPHARM-PDM (https://www.nitrc.org/projects/spharm-pdm)

To preserve the anatomy of both cortex and subcortical structures, their surfaces were inflated, parameterized to a sphere, and registered to a spherical surface template using a rigid body registration (Besson et al., 2014; Azcona et al., 2020). Then, the surface templates were converted to triangular meshes following a triangulation scheme. Here, each edge was assigned a weight $\mathbf{e}_{i,j}$ which corresponds to its geodesic distance $\psi_{i,j}$ along the surface, such that:

$$\mathbf{e}_{i,j} = \frac{1}{\sigma\sqrt{2\pi}} \exp \left\{ -\frac{1}{2} \left( \frac{\psi_{i,j}}{\sigma} \right)^2 \right\} \tag{6}$$

### B.2. Data Statistics

4072 T1-weighted MRI volumes are used in this paper, which corresponds to 492 subjects. Basic demographic information includes the number of samples with different labels, as well as the age and gender distribution of the subjects are shown in the Table below:

Table 4: Demographic information of the ADNI dataset, including sample numbers across different labels and the age and gender distribution of the subjects.

|      | Subjects | Samples | Gender (M/F) | Age (mean$\pm$ SD) | MMSE (mean$\pm$ SD) |
|------|----------|---------|--------------|--------------------|---------------------|
| EMCI | 259      | 1880    | 143/116      | $71.0 \pm 7.6$     | $27.4 \pm 1.4$      |
| LMCI | 233      | 2156    | 127/105      | $74.1 \pm 7.1$     | $25.3 \pm 2.4$      |

[1] *MMSE* Mini-mental State Examination

## Appendix C.  Other Supplementary Materials

### C.1.  Supplementary Materials for Cost-Oriented Sampling

#### C.1.1. Explanation of Parameter Selection

The parameter $n$ is directly associated with the input size of the fully connected layer, i.e., $input\_size = n \times channel\_size$. Through experiments, a large $n$ (i.e., $n = 256$) does not significantly improve the final classification results but rather wastes a lot of computation resources. On the contrary, a small value of $n$ (e.g., $n = 64$) will make every pooling region too large, resulting in the network's inability to recognize subtle features of the mesh surface. In our experiment, $n$ is empirically determined as 128.

For the number of sampling, $|\mathcal{V}_i^d|$, we fine-tune this parameter starting from 60, which is half of the average number of vertices in each pooled region ($14848/128 = 116$). This parameter does not significantly affect the result until it approaches 116 (i.e., averaging all vertex embeddings within the pooling regions) or 0 (i.e., ignoring the information of all decimated vertices). As a result, the number of sampling is empirically determined as 40.

#### C.1.2. Simulation of Sampling

In this section, we provide a toy example of applying oriented sampling on a randomly picked pooling region. In keeping with the experiment settings in our paper, the mesh surface is divided into $n = 128$ pooled regions, and $|\mathcal{V}_i^d| = 40$ decimated vertices are sampled from each region.

To simulate the training process, we randomly pick a pooling region $r_i$ on the right thalamus surface containing 88 vertices. Then, we apply our oriented sampling with three different sampling numbers, i.e., 10, 40 (our experiment setting), and 80 for 1000 times.

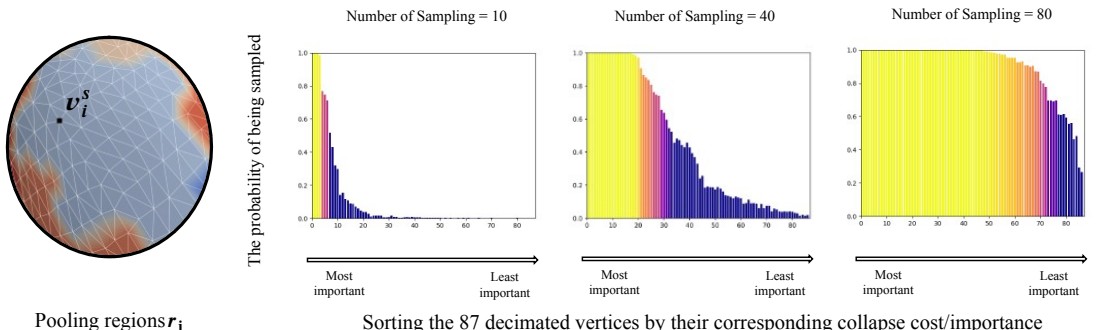

Figure 5: A toy example of applying oriented sampling on pooling region $r_i$ for 1000 times, using 3 different sampling numbers

As shown in the figure 5, when the number of samples is 40, the top 20 vertices with the highest collapse cost/geometric importance will continuously be sampled and participate in the information propagation, which ensures the overall stability of the results.

### C.2. Supplementary Materials for Table 2(a)

This section will provide supplementary materials for the baseline experiment listed in Table 2(a).

- MESHSIM (Ranjan et al., 2018) adopts the mesh simplification process in Sec. 2.3.1 to extract hierarchical features from the mesh surface.

In order to avoid "information loss", we applied the global pooling module individually to each of the 14 separate subcortical structures, and subsequently concatenated the results for classification. The implementation of our baseline experiments in the code was done using *Pytorch Geometric 2.0.4 Doc* (Fey, 2022), with default parameter settings unless otherwise stated.

- GLOBAL MEAN/ADD POOLING (Lin et al., 2013) return batch-wise graph-level-outputs by averaging/adding features across the vertex dimension. (Note that Grad-CAM does not support Global Max Pooling, resulting in uninterpretable experiment results. We decide not to include it in the baseline comparison here).

- GLOBAL ATTENTION POOLING (Li et al., 2015) leverage soft attention mechanism to decide which vertices are relevant to the current graph-level task, which can be expressed as:

$$\mathbf{r}_i = \sum_{n=1}^{N_i} \mathrm{softmax}\left(h_{\mathrm{gate}}\left(\mathbf{x}_n\right)\right) \odot h_\Theta\left(\mathbf{x}_n\right)$$

  In our experiment, $h_{\mathrm{gate}}$ is chosen to be a 3 layers multi-layer perceptron, with channel size $[16, 8, 1]$.

- SORTPOOL (Zhang et al., 2018) sorts the node embedding in ascending order and selects the features of top-k nodes. In our experiment, the number of nodes to hold for each graph, $k$, is chosen to be 1.

- SET2SET (Vinyals et al., 2015) adopts iterative content-based attention to generate an order invariant representation. In our experiment, the number of nodes to hold for each graph, $k$, is chosen to be 1.

- GMT (Baek et al., 2021) clusters nodes of the entire graph via attention-based pooling operation. The parameter *in_channel*, *hidden_channel*, and *out_channel* are respectively chosen to be 16, 8, 1

