# OpenReview forum: "Stage Detection of Mild Cognitive Impairment: Region-dependent Graph Representation Learning on Brain Morphable Meshes"
_MIDL.io/2023/Conference — MIDL 2023 Poster_

### Official Review · Reviewer_sTfY · 2023-01-28

**Confidence:** 4
**Preliminary Rating:** 3
**Recommendation:** Poster

**Summary:**

The paper at hand addresses the problem of MCI sub-type classification from MRI-based meshes. The authors employ a GNN with a newly proposed regional pooling method. The authors use a public dataset of 492 subjects to train and evaluate their method. Comparisons to other pooling operations are performed and a CAM-based visualization is shown.

**Strengths:**

- The paper is well-written and easy to follow
- The authors propose a new GNN building block that may improve performance for MCI sub-type classification
- The authors compare their approach against other methods
- The authors provide CAM results that allow for some interpretability of the model’s behavior


**Weaknesses:**

- Motivation: the authors state that a major motivation for EMCI vs LMCI classification is the inaccuracy of current methods. However, it is not explained what the ground-truth of the employed dataset is based on. Was the GT based on these inaccurate tests? The authors should briefly comment on this and the implications for their approach
- Motivation: the authors state that MCI sub-type classification has recently gained interest in the community. However, it is not entirely clear how classification of these subtypes would help in a clinical setting (e.g. different treatment paths? What are the consequences of incorrect sub-type classification?). A few words on this would be helpful

- Method: the authors compare their method to a selected number of pooling methods. The authors should explain why they selected this subset of methods to compare to, and not others that are available, see e.g. the available implementations of several papers in [1]

- Table 1: the authors should state what the standard deviation refers to. Variation across folds? Also, macro averaging is arguably misleading, as it does not take class imbalance into account. Metrics per class would more helpful in interpreting the results.
- The data splitting/experimental setup for the results in table 2 a) is unclear. It sounds like the authors selected one fold from the previous experiment and then repeatedly trained on the single fold. This needs to be clarified and there should be a reasoning why the setting was changed from the previous experiment.
- Confidence intervals, standard deviations, or statistical tests would help in interpreting the results in Table 2a) – without, it is not clear whether there is a substantial gain by using the authors’ proposed method
- Table 2b) – the value of this table is very unclear. Comparing completely different datasets with completely different methods is not very interpretable. Also, the authors claim the superiority of their approach based on this table while it is not clear what this refers to. The highest accuracy is scored by a different method. Also, the accuracies cannot be compared since they all have completely different datasets.
- The authors do not seem to differentiate between validation and tests data which might skew the results

[1] https://pytorch-geometric.readthedocs.io/en/latest/modules/nn.html#pooling-layers

**Deanonymize Review:**

yes

**Detailed Comments:**

See weaknesses

**Paper Type:**

methodological development

**Questions To Address In The Rebuttal:**

The authors should address the main points mentioned in the weaknesses. This includes clarifying the data splitting approaches and experimental settings, the purpose of table 2b, the overall motivation, as well as the method selection.

---

### Official Review · Reviewer_d9pX · 2023-02-01

**Confidence:** 4
**Preliminary Rating:** 4
**Recommendation:** Poster

**Summary:**

The paper presents a graph neural network (GNN) based approach for Mild Cognitive Impairment disease stage detection. Specifically, they use popular Graph Attention Convolution with their proposed REGIONPOOL operation to create a GNN for disease stage classification. They apply it to triangular meshes of both cortical and subcortical structures. The results demonstrate that their methods gives better performance compared to previously published non-deep learning methods. Similarly, they also show improved performance compared to some of the other graph pooling methods. Their Grad-CAM maps show that their method focuses on known AD biomarkers. Overall, the methods sound novel with satisfactory experimental and result section.

**Strengths:**

* Good introduction regarding clinical usefulness of the disease stage classification between LMCI and EMCI.
* Overall REGIONPOOL method is well defined and clearly written.
* Use of Grad-CAM is commendable for verifying the learned region of interest.
* Experiments on large-scale publicly available dataset.

**Weaknesses:**

* While I appreciate the good introduction to the clinical usefulness of the task, it is missing a key literature review section on graph neural networks in medical image analysis. There has been a lot of work done in this area for both the general application of GNN in medical image analysis and specific applications in AD classification. Authors should write a small section (atleast one paragraph) on this. Relevant literature review paper [1].
* Many necessary parts are described in the appendix rather than in the main paper. Authors refer to the appendix in section 2.2 and 2.3, and then writing is done such that it is assumed that the readers have read the appendix. While I understand that authors need to adhere to a page limit, the writing can be improved. For example, as authors are using an already published graph attention network, they can move section 2.2 to the appendix and bring a write-up related to section 2.3 to the main paper from the appendix. This would be much easier to follow as the main contribution of the paper is defined in section 2.3
* It is not clear how differences in input-feature length between cortical and sub-cortical meshes are handled. For example, cortical meshes have 6-dimensional features while sub-cortical have 3-dimensional features. Do the authors have different branches for cortical and sub-cortical meshes? If that is the case, it is not clear either in Figure 3 or in the paper write-up. If that is not the case, then authors may want to clarify that.
* The authors use a total of 4072 T1 MR images from a total of 492 patients. This turns out to be approximately 8 T1 MR images per patient. Do authors use longitudinal data for this or data from only a single time point? If they use longitudinal data, then disease stage diagnosis might have changed between these data. If that is not the case and it is from a single time point, then I am not sure how having 8 MR images for a single patient would be useful as they give exactly the same information after registration. A clarification regarding the same would be useful.
* In section:3.2, I am unsure why the authors have decided on n=128 pooling region and 40 decimated vertices. Was it empirically decided? In that case, did the authors see a drastic change in performance with a change in these parameters?  Maybe the authors want to provide these results in the appendix.
* If I understood correctly, the authors say that one of the contributions of their work is including decimated vertices in the pooling operation in addition to the selected vertices. An ablation study regarding the usefulness of including these decimated vertices would be useful.
* Authors only compare their results against MESHSIM, GAP, SET2SET, and SORTPOOL. There have been a lot of already publicly available graph pooling methods in PyTorch Geometric [2]. Authors may want to include some of them in their work.
* Similarly, the authors are missing relevant work [3], which developed a learnable pooling method for brain surface analysis and experimented on AD disease stage classification. Authors should compare their method against this work as the code is already publicly available.
* Also, there has been a cluster-based pooling method in CV [4]. Authors may want to discuss similarities and differences between this work and their proposed method.

[1] Ahmedt-Aristizabal, D., Armin, M.A., Denman, S., Fookes, C. and Petersson, L., 2021. Graph-based deep learning for medical diagnosis and analysis: past, present and future. Sensors, 21(14), p.4758.

[2] https://pytorch-geometric.readthedocs.io/en/latest/modules/nn.html#pooling-layers

[3] Gopinath, K., Desrosiers, C. and Lombaert, H., 2020. Learnable pooling in graph convolutional networks for brain surface analysis. IEEE Transactions on Pattern Analysis and Machine Intelligence, 44(2), pp.864-876.

[4] Wang, C., Pelillo, M. and Siddiqi, K., 2019. Dominant set clustering and pooling for multi-view 3d object recognition. arXiv preprint arXiv:1906.01592.

**Deanonymize Review:**

no

**Detailed Comments:**

* Authors write Tondellie et al. (Tondelli et al., 2012). I think the authors are using \cite{} command here. If they replace it with \citet{}, it will remove repetition and lead to a single Tondellie et al., 2012
* Some symbols are directly introduced in the paper without mentioning what they mean. For example, $h_i^{(k)}$. It is not mentioned in the main paper text what this symbol represents. Similar comment for $[x_i,  y_i,  z_i,  1]^T$
* There is a typo in line-5, page-6. "Besides, to get fully used of the inherent coarse-grained hierarchies of the brain meshes, ...." should be "Besides, to get full use of the inherent coarse-grained hierarchies of the brain meshes, ..."
* GAP introduced in section:3.3 should be written as Global Average Pooling and have a relevant citation [1].
* Section 3:3 has inconsistency in tenses between sentences. I think line-8 should be written as "All experiments were conducted ..." instead of "All experiments will be conducted ..." as the rest of the paragraph is written in the past tense.
* Title of section 3.4 should have "Grad-CAM" instead of "Grad-cam".

[1] Lin, M., Chen, Q. and Yan, S., 2013. Network in network. arXiv preprint arXiv:1312.4400.

**Paper Type:**

both

**Questions To Address In The Rebuttal:**

All the points in the Weakness section apart from point-7. Also, it would be nice if the authors could incorporate some of the suggestions in the detailed comments section, as it will improve the paper's readability.

Edit: Authors have provided a good response to review during the rebuttal period. I am happy to increase my rating to weak accept.

---

### Official Review · Reviewer_MeCR · 2023-02-02

**Confidence:** 4
**Preliminary Rating:** 2

**Summary:**

To authors claim that brain shape is key factor in distinguishing early stage mild cognitive impairment (EMCI) vs late state cognitive impairment (LMCI). In order to take advantage of this, the authors construct shape meshes of from brain MRIs and train a graph attention network to classify the meshes as belonging to patients with LMCI or EMCI. The authors introduce a new pooling method (RegionPool) which uses quadric error to first rank vertices and then sample the vertices to be decimated weighted by on this ranking.

**Strengths:**

- The paper shows promising results using this new pooling method. When compared to the baseline methods RegionPool out performs all other methods across the board in precision, recall, F1, PR-AUCs, and ROC-AUC


**Weaknesses:**

- The paper lacks details on related work which places the proposed method in context, distinguishing the difference between existing methods and the proposed.
- The paper would benefit from more experimental results or analysis into why RegionPool is a better candidate than other, simpler methods of pooling.

**Deanonymize Review:**

no

**Detailed Comments:**

"specially computed" - typo?
"theinherent.." typo, missing space
"Predicted possibility" - typo?

**Paper Type:**

methodological development

**Questions To Address In The Rebuttal:**

- Could the authors elaborate on the claim that "the computational complexity of the simplification process, which makes the in-network mesh simplification process always deterministic. Such a static scheme ignores the information encoded by those decimated
vertices..." - why does complexity inherently lead to determinism and why is this one of the two major drawbacks? Does the averaging of the vertex representations of decimated vertices in (Yuan et al., 2020) not propagate information through the network?
- Why is table 1 constructed using cross-fold validation while Table 2 isn't? Would picking a fold to do further evaluation based on the "highest accuracy" bias the results?
- What is the key difference between the proposed method and that of (Ranjan et al., 2018) which also uses quadric error to subsample meshes, other than the sampling of vertices? What is the motivation for the sampling? What benefits does this bring?

---

### Official Review · Reviewer_T3pJ · 2023-02-03

**Confidence:** 4
**Preliminary Rating:** 3
**Recommendation:** Poster

**Summary:**

This paper introduces a novel deep learning method based on region specific brain meshes. The model uses a GNN to extract embeddings from brain regional meshes which are then simplified through region pooling. Finally mesh embeddings are extracted and fed to a FCNN for prediction. The authors tested their approach on MCI type classification using the ADNI database. Their results are better than when using other graph pooling existing methods but classic machine learning methods (not using meshes) achieve better classification accuracy.

**Strengths:**

The paper introduces interesting methods and a novel pipeline. Each step of the pipeline is described in details. I think the the approach could be of interest for the MIDL community and useful in many applications, especially with the explainability (grad cam) component.

**Weaknesses:**

I found the paper difficult to read and follow as:

•	The motivation is not clear. The introduction is very focused on the clinical task, however I would rather remove some of this and add a related work section were similar models would be described (= models presented in Table 2 (a)) and explain the novelty of the method compared to those.

•	Too many notations are introduced. Some of them are not used later on and could be removed (A and R in section 2.1).

•	The ordering of the Methods section is weird. I would start with the overview of the pipeline (Figure 3), name the pipeline (HET-BMAT), and then go into the details of each part.

•	The paper needs a second read as they are several typos/grammatical errors (see minor comments)


**Deanonymize Review:**

no

**Paper Type:**

both

**Questions To Address In The Rebuttal:**

Major:

* See weaknesses section

* In table 2 (b), is there an existing classic CNN trying to solve that task? That would give us an idea if using meshes/brain shape is better than using image intensity for this example task.

* Section 2.3.2: It is not very clear how patches are chosen. Is it random? And if yes, I am wondering how much the variability in patches between model runs affect the results.

Minor:

* Some parts of the pipeline are based on previous work, which is fine, but it is hard to figure out which methods are new and which ones come from the literature.

* Figure 1 could be removed to save some space as it introduces notations that are not used later and we already see an example of original brain mesh on Figure 2. On Figure 1, Rc and Rs notations are missing. On the second panel it should be “Subcortical connection”.

* Figure 2 (e): it is very difficult to see color differences.

* Table 4: What does n=259 represent? It is different from the number of samples

* P2, line 6: decreasing OF 15-40%

* P5, line 19: choice should be choose

* P6, line 5: theinherent missing space

---

### Meta-Review · Area_Chair_Dnmf · 2023-02-24

**Recommendation:** Accept (Poster)
**Confidence:** 5

**Metareview:**

This paper presents a novel and interesting approach to AD subtype identification via brain structure shape analysis using graph learning with region-dependent pooling. While the proposed method has novelty, all reviewers raised questions about relevant studies, which are lacking in the paper. Reviewers also provided detailed suggestions for clarifying the method section and improving validation. The rebuttal and the revised paper intensively (but not all) addressed those critiques.